# L-Fucose-Rich Sulfated Glycans from Edible Brown Seaweed: A Promising Functional Food for Obesity and Energy Expenditure Improvement

**DOI:** 10.3390/ijms25179738

**Published:** 2024-09-09

**Authors:** Jimin Hyun, Hyo-Geun Lee, Jun-Geon Je, Yun-Sang Choi, Kyung-Mo Song, Tae-Kyung Kim, Bomi Ryu, Min-Cheol Kang, You-Jin Jeon

**Affiliations:** 1Major of Food Science and Nutrition, Pukyong National University, Busan 48513, Republic of Korea; 2Department of Marine Life Sciences, Jeju National University, Jeju 63243, Republic of Korea; 3Research Group of Food Processing, Korea Food Research Institute, Wanju 55365, Republic of Korea; 4Department of Food Science & Biotechnology, Sungshin Women’s University, Seoul 01133, Republic of Korea

**Keywords:** brown seaweed, *Scytosiphon lomentaria*, sulfated glycans, fucoidan, metabolic disorders, energy expenditure, AMPK, zebrafish

## Abstract

The global obesity epidemic, exacerbated by the sedentary lifestyle fostered by the COVID-19 pandemic, presents a growing socioeconomic burden due to decreased physical activity and increased morbidity. Current obesity treatments show promise, but they often come with expensive medications, frequent injections, and potential side effects, with limited success in improving obesity through increased energy expenditure. This study explores the potential of a refined sulfated polysaccharide (SPSL), derived from the brown seaweed *Scytosiphon lomentaria* (SL), as a safe and effective anti-obesity treatment by promoting energy expenditure. Chemical characterization revealed that SPSL, rich in sulfate and L-fucose content, comprises nine distinct sulfated glycan structures. In vitro analysis demonstrated potent anti-lipogenic properties in adipocytes, mediated by the downregulation of key adipogenic modulators, including 5′ adenosine monophosphate-activated protein kinase (AMPK) and peroxisome proliferator-activated receptor γ (PPARγ) pathways. Inhibiting AMPK attenuated the anti-adipogenic effects of SPSL, confirming its involvement in the mechanism of action. Furthermore, in vivo studies using zebrafish models showed that SPSL increased energy expenditure and reduced lipid accumulation. These findings collectively highlight the therapeutic potential of SPSL as a functional food ingredient for mitigating obesity-related metabolic dysregulation by promoting energy expenditure. Further mechanistic and preclinical investigations are warranted to fully elucidate its mode of action and evaluate its efficacy in obesity management, potentially offering a novel, natural therapeutic avenue for this global health concern.

## 1. Introduction

The COVID-19 pandemic, declared by the World Health Organization (WHO), has profoundly impacted global health. The resulting decline in social activities and the rise of contactless interactions have contributed to a surge in metabolic diseases, particularly obesity [1,2]. Defined as a body mass index (BMI) exceeding 30 kg/m^2^, obesity poses a significant threat to public health, with rising prevalence rates worldwide [3,4,5]. The transition back to pre-pandemic norms, coupled with pervasive sedentary lifestyles and increased reliance on contactless interactions, has exacerbated this issue, increasing the socioeconomic burden associated with obesity [6].

Obesity treatment encompasses a variety of interventions, including dietary modifications, exercise regimens, pharmacotherapy, and surgical procedures. However, the frequent occurrence of adverse effects and challenges in maintaining consistent adherence necessitate the exploration of safer and more efficacious alternatives [7]. Individual effort and concerns regarding adverse effects significantly influence the treatment process, and patient compliance can fluctuate depending on the selected intervention [7]. Recently, dietary interventions targeting energy balance regulation have garnered attention as a promising avenue for obesity management [8]. This approach presents a viable method for achieving effective weight reduction while mitigating the risk of adverse events, and ongoing research endeavors are actively investigating its potential.

Owing to its rapid expansion, the global health functional food (HFF) market has become an integral part of the food industry. In particular, South Korea has built high consumer confidence in HFFs based on significant physiological effects demonstrated through systematic research, showing promise as a country that has achieved >40% HFF market growth over the past four years [9,10]. Research on HFF candidate materials suitable for industrialization for continuous market growth is required, and the importance of oceans with their variety of available resources and abundant biomass is highlighted [11].

Marine macroalgae are exposed to various mechanical stresses and risks of predators in natural ecosystems [12]. Numerous types of polysaccharides in algae are highly abundant and confer resistance to dangers from environmental stress and predatory interactions, such as the skeletal structure in algal cell walls [12]. In particular, polysaccharides isolated from brown algae exhibit unique structural properties and act as a carbon pool in oceanic circumstances [13]. Moreover, polysaccharides originating from brown algae are an edible substance with great potential, which contains physiologically active substances that show excellent effects on metabolic diseases and immune regulation [14]. One of the major components of brown seaweed is L-fucose-containing sulfated polysaccharides (FCSPs), a type of sulfated polysaccharide characterized by a glycan structure with a backbone composed of [Fuc_2_(SO_3_)]^−^ [14]. However, the chemical structure of FCSPs varies in different brown seaweed species according to the monosaccharide composition, number of sulfate groups, position of the sulfate residue in the structure, and molecular weight [15]. Intriguingly, the biological effects of FCSPs, including anti-lipogenic and anti-inflammatory effects, largely depend on the sulfate content, regardless of the brown seaweed species [16,17,18]. Therefore, elucidating the composition and glycan structures of FCSPs is crucial for understanding the unique structural characteristics of seaweed-specific FCSPs and their associated bioactivities, ultimately facilitating their industrial application.

Adipocytes are a major source of energy produced through lipogenesis using overproduced ATP in organisms; they are considered an important counterstrategy for obesity control [19]. An increased AMP:ATP ratio activates AMP-activated protein kinase (AMPK) and promotes lipolysis, the inhibitor of lipogenesis and glucose uptake, and energy expenditure accompanied by mitochondrial ox-redox cycle enhancement in adipocytes [20]. AMPK is a key factor involved in regulating fat accumulation and energy metabolism through dietary interventions, and it is being actively researched as a potential target for obesity treatment [21,22,23]. *Scytosiphon lomentaria* (Lyngbye) Link, nom. Cons. 1833 (SL) is an atypical lobed brown algae species, commonly native to the low-tide area of Jeju Island. In addition, it is widely distributed along the west coast of the North Pacific Ocean, Europe, and East Asia, including China, Japan, and the southern coast of Korea, showing abundant biodiversity worldwide [24,25,26,27,28]. SL is expected to contain excellent functional substances, including fucoidan, but related research is lacking [29]. An ethanol extract from SL showed excellent muscle growth promotion through myostatin inhibition in vitro and alleviated hyperlipidemia by reducing plasma triglyceride levels in a diet-induced zebrafish obesity model [30]. However, which functional ingredients in SL release metabolic burden and their underlying molecular mechanisms remain to be discovered. In this study, the physicochemical features and potent anti-obesity effects of a refined polysaccharide fraction isolated from SL were investigated.

## 2. Results

### 2.1. Anti-Lipid Accumulation Effect of Polysaccharides Extracting Enzymatic SL Hydrolysates

Enzymatic hydrolysis is widely used as a natural product preparation tool to screen novel substances with multiple bioactivities [31]. SL was enzymatically hydrolyzed using five different polysaccharide-targeting enzymes under optimal working conditions to extract the polysaccharide contents. These enzymatic SL plus distilled water (vehicle) hydrolysates showed a significant increase in extraction yield compared to the vehicle-hydrolysate-only group (Figure 1A). Subsequently, enzymatic SL with vehicle hydrolysates was applied to 3T3-L1 cells, murine pre-adipocytes undergoing differentiation into adipocytes (Figure 1A). ORO staining revealed a distinct crimson intensity in the vehicle-treated- fully differentiated group of 3T3-L1 cells, compared with the non-treated group (Figure 1A). However, this pattern was significantly reduced by the 200 µg/mL of the six enzymatic SL hydrolysates (Figure 1A,B). All enzymatic SL hydrolysates showed remarkable retardation of lipid accumulation without any significant cytotoxicity, and SLC showed the highest inhibitory effect on lipid accumulation in fully differentiated 3T3-L1 adipocytes (Figure 1A–C). Thus, SLC was selected for the subsequent polysaccharide fraction refinement step based on its outstanding lipogenesis inhibition, despite its relatively lower extract yield value.

### 2.2. Characterization of Sulfated Polysaccharide Fraction Isolated from SL via Ethanol Precipitation

The isolation of polysaccharides binding with sulfated groups from brown seaweed extract by high-concentration ethanol precipitation at low temperatures is a simple and cost-effective process [32]. Thus, sulfated polysaccharide fractions isolated from SL via ethanol precipitation (SPSL) characteristics were assessed for proximate composition and monosaccharide constituents (Table 1 and Figure 2A). The polysaccharide content was distinctly increased in SPSL (41.99%), although proteins were also co-extracted (12.10%) (Table 1). Interestingly, L-fucose was the most abundant neutral monosaccharide (43.17%), followed by L-galactose (28.71%), xylose (17.49%), D-glucose (8.12%), L-rhamnose (1.66%), D-fructose (0.69%), and L-arabinose (0.16%) in SPSL, and the amount of L-fucose was clearly higher than that in SLC (33.68%) (Figure 2A). As expected, sulfate content also dramatically elevated in SPSL (Figure 2B). Compared to the standardized polysaccharides, SPSL exhibited a proximate molecular weight which is concentrated within the range of ≥60–500 kDa (Figure 2C). To monitor the structural bonds in SPSL without destruction, SPSL was analyzed using FT-IR spectra to compare their structural properties to the standard fucoidan isolated from *Undaria pinnatifida* within the 500–2000 cm^-1^ wave range (Figure 2D). In comparing the three specimens, the marked stretching of the absorption peaks was displayed at 845 cm^−1^ (C-O-S), 1025 cm^−1^ (C-O-C), 1220–1270 cm^−1^ (S=O), 1404 cm^−1^ (C-OH), and 1640–1650 cm^−1^ (C=O), respectively (Figure 2D). Intriguingly, as the number of polysaccharides and sulfate content escalated in SPSL, the 845 cm^−1^ and 1220–1270 cm^−1^ spectra which indicate sulfate groups, became similarly aligned with the fucoidan standard (Figure 2D). Although there was no significant difference in the molecular weight range, high L-fucose and sulfate content were observed in SPSL, and its structural features were confirmed to be similar to commercially available purified fucoidan. Therefore, analysis of the major glycans constituting SPSL was performed. UPLC-Q-TOF-MS analysis of acid-hydrolyzed SPSL, using our glycan library for structure identification, revealed several peaks indicating specific substances (Figure 2E,F). Glycan library matching identified nine types of glycan structures: (1) [Fuc_2_Hex(SO_3_)_2_]^2−^ (630.08 g·mol^−1^); (2) [Fuc_2_Hex_2_(SO_3_)_2_]^2−^ (646.07 g·mol^−1^); (3) [Fuc(SO_3_)]^−^ (243.02 g·mol^−1^); (4) [Fuc_4_(SO_3_)_2_]^2−^ (760.14 g·mol^−1^); (5) [Fuc_2_(SO_3_)]^−^ (389.08 g·mol^−1^); (6) [Fuc_6_Hex(SO_3_)_3_]^3−^ (1293.26 g·mol^−1^); (7) [Xyl_2_(SO_3_)_2_]^2−^ (439.99 g·mol^−1^); (8) [FucXyl_3_(SO_3_)_5_]^5−^ (954.94 g·mol^−1^); and (9) [FucXyl(SO_3_)]^−^ (375.06 g·mol^−1^) (Table 2). Among these, [Fuc_2_(SO_3_)]^−^, representing the major backbone of fucoidan, was the most abundant glycan structure in SPSL (69.38%) (Figure 2E and Table 2). These results establish SPSL as a sulfated polysaccharide fraction closely resembling refined fucoidan in its features.

### 2.3. Anti-Lipogenesis Effect of SPSL in 3T3-L1 Adipocytes

Subsequently, SPSL was used to measure cytotoxicity before being applied to adipogenic differentiation to determine the correlation between changes in sulfate content and the regulation of adipogenesis in 3T3-L1 adipocytes (Figure 3A). A series of SPSL concentrations did not cause any toxicity in fully differentiated 3T3-L1 cells. Similar to SLC (200 µg/mL), SPSL also manifested outstanding inhibition of adipogenesis in murine adipocytes at lower doses (50, 25, and 12.5 µg/mL) (Figure 3B,C). These results suggest that the anti-lipogenic effect of SPSL is highly dependent on sulfate content in vitro.

### 2.4. SPSL Blunted Adipogenesis by Regulating Key Adipogenic Modulators

Adipogenesis is orchestrated by the complex regulation of post-transcriptional changes as well as gene expression programs [33]. SPSL decreased the protein expression of PPARγ and C/EBPα in a dose-dependent manner, compared to fully differentiated cells. In addition, we observed excellent inhibition of key regulators of lipogenesis, such as SREBP-1 and ATGL, target molecules of PPARγ as a master transcription factor (Figure 4A–E) [34]. In addition, SPSL upregulated AMPK phosphorylation in a dose-dependent manner (Figure 4A,F). AMPK activation by phosphorylation also led to the phosphorylation of ACC, the major target molecule of AMPK activation, following the increased SPSL dose gradient (Figure 4A,G). In conclusion, the distinct anti-lipogenic effect of SPSL was observed through the protein expression changes of essential adipogenesis regulators controlling the adipogenic master regulator and its co-factors.

### 2.5. Inhibiting AMPK Activation Disturbed the Anti-Lipogenic Effect of SPSL

AMPK activation through phosphorylation can lead to excessive energy storage as fat under obese conditions [35]. AMPK plays a pivotal role in fat formulation as well as in the metabolic catabolism and anabolism cycle as a central energy sensor [35]. SPSL application to adipocytes during differentiation showed a dramatic dose-dependent activation of the AMPK mechanism, along with decreased fat accumulation in 3T3-L1 adipocytes, compared to the control group (Figure 4). Thus, SPSL and Compound C, an AMPK inhibitor, were co-administered during adipocyte differentiation to confirm whether AMPK inhibition blunts the anti-lipogenic effect of SPSL (Figure 5). Compound C treatment slightly decreased lipid accumulation, compared to the media for the differentiation-induction-alone group, but the difference was not significant (Figure 5A,B). Consistently, SPSL inhibited adipogenesis, which was restored by blocking AMPK activation through Compound C co-treatment (Figure 5A,B). These results suggest that inhibition of lipid accumulation in SPSL is maintained through AMPK activation.

### 2.6. SPSL Enhanced the Energy Expenditure Rate and Lipid Accumulation in Zebrafish

AMPK activation is the most promising treatment target for obesity because of its protective role in diet-induced obesity exerted via augmentation of the systemic energy expenditure rate [36]. This enhanced energy expenditure rate is probably due to the improved oxygen consumption rate in white adipose tissue [37]. Based on this knowledge, 2 dpf zebrafish larvae were incubated in 1% Alamar Blue with a series of SPSL doses to assess the energy expenditure rate of NADH_2_ formulation (Figure 6A). As the positive control, insulin induced a color change toward pink, which indicated remarkably increased energy expenditure, compared with the blank group (Figure 6B). Interestingly, the lowest SPSL dose significantly increased the energy expenditure rate to the same extent as the positive control group. Increasing doses further increased the rate by approximately 2.5–3.0-fold, compared to the blank (Figure 6B). In addition, these dose-related treatments did not induce toxicity in the zebrafish larvae (Figure 6C).

Since the remarkable energy expenditure elevation by SPSL supplementation to zebrafish, it was co-introduced to the zebrafish to measure whether SPSL may reduce the lipid accumulation in zebrafish that fed a high cholesterol supplementation (Figure 6D). The cholesterol supplementation only using egg yolk clearly upregulated a lipid accumulation in the zebrafish larvae blank group (Figure 6D,E). The whole-body triglyceride level also significantly rose in the blank group (Figure 6F). On the other hand, SPSL treatment outstandingly blunted lipid accumulation as well as the triglyceride level in a zebrafish dose-dependent manner (Figure 6D–F). In addition, cholesterol supplementation in zebrafish markedly increased lipid deposition in the dorsal aorta, one of the major circulatory systems (Figure 6D). However, SPSL treatment distinctly lowered the high level of lipid in the circulatory system in a dose-related pattern (Figure 6D). Altogether, the AMPK activation-based mechanism of the anti-lipogenic effect of SPSL in vitro also affected energy expenditure enhancement in zebrafish in vivo. Thus, these results suggest that a supply of SPSL may inhibit the accumulation of lipids in the whole body by increasing the energy consumption rate and simultaneously controlling hyperlipidemia to help prevent metabolic diseases.

## 3. Discussion

Proper maintenance of BMI is essential for preventing various metabolic disorders associated with obesity. The rise of sedentary lifestyles, exacerbated by the COVID-19 pandemic, has led to a surge in obesity and diabetes [6]. Consequently, public interest in natural product-derived HFFs with fewer side effects and clear improvements, compared to conventional drugs, has fueled interest in exploring the potential of marine-derived biomolecules. Sulfated polysaccharides offer a potential mechanism for mitigating metabolic stress, including ectopic fat accumulation, obesity, and diabetes. This is achieved by inhibiting adipogenesis in white adipose tissue and the liver through enhanced glucose uptake via AMPK activation in hepatocytes and adipocytes [37,38,39,40]. Therefore, elucidating the physiological activity of HFFs could enhance their industrial application as food agents for weight control and mitigating obesity-related metabolic side effects [41]. Despite its widespread use as a marine-sourced edible ingredient due to its high biomass, SL has been reported to possess a limited range of biological activities [42]. Although SL reportedly contains fucoidan, a sulfated polysaccharide, its associated biological effects remain unexplored [43]. In this study, SPSL not only contained a higher level of polysaccharides than SLC (Table 1 and Figure 2B) but also exhibited the highest amount of L-fucose and the highest prevalence of sulfated glycans (Figure 2A,E).

The existing literature suggests that sulfated polysaccharides with anti-obesity effects in vivo typically exhibit average molecular weights between 5–500 kDa [44,45,46]. Interestingly, SPSL exhibited a molecular weight range primarily under 500 kDa, with some components falling below 60 kDa (Figure 2C). Despite potential limitations in biomolecular permeability due to its large molecular size, highly polymerized structures may induce satiety and extended gastric emptying through mechanisms such as gastric renal receptor stimulation and delayed nutrient absorption [47]. However, the exact mechanism of algal sulfated polysaccharides remains unclear, their abundance of sulfate groups has been linked to high water solubility compared to general dietary fibers presenting gel-forming properties [48]. Therefore, the presence of sulfate groups is a key factor distinguishing SPSL from many other dietary fibers in terms of solubility, and this enhanced solubility may indeed contribute to its potentially higher bioavailability. In the FT-IR spectra, SPSL shared several functional groups with commercial fucoidan, such as C=O, C-O-C, S=O, and C-O-S bonds (Figure 2C), consistent with previous studies demonstrating the anti-obesity effects of fucoidan isolated from brown algae [39,49]. As revealed by the analysis of specific sulfate group-linked glycan structures present in SPSL, the major polysaccharides in SPSL are suggested to have a backbone primarily composed of sulfated fucose and xylose (Figure 2E). These results are distinctly aligned with the monosaccharide constituents in SPSL, which showed an abundance of L-fucose and xylose contents (Figure 2A). Sulfate groups in sulfated polysaccharides, primarily composed of fucoidan, refer to the diverse positions where these groups attach to algal monosaccharides. L-Fucose, the main monosaccharide in fucoidan, can have sulfate groups attached at various positions, such as the 2nd, 3rd, or 4th carbon [15,17]. This diversity in sulfate group attachment sites creates structural variations in fucoidan, potentially influencing its biological functions, including antioxidant, anti-inflammatory, and anti-cancer effects [32]. Numerous studies have revealed the relationship between fucoidan’s structure and specific physiological activities [15,38,44]. While the precise positional elucidation of sulfate isomers within SPSL remains challenging, identifying sulfate groups associated with specific monosaccharides and the fundamental structural unit in SPSL, which exhibits remarkable anti-obesity effects, lays a foundation for future investigations. In summary, notwithstanding its high molecular weight, the abundance of sulfated glycan units within SPSL suggests it possesses favorable aqueous solubility, facilitating its transit through the gastrointestinal tract. Subsequent utilization by digestive enzymes or the gut microbiome may yield diverse sulfate isomers, potentially eliciting a range of physiological effects, including anti-obesity activity. Further investigation is warranted to elucidate the precise mechanisms and therapeutic implications of these potential bioactivities.

AMPK, a key regulator of energy metabolism, interacts with PPARγ, a master regulator of adipose tissue [50]. PPARγ and its co-regulator C/EBPα are upregulated during adipogenesis in adipocytes, [34] as confirmed in fully differentiated 3T3-L1 adipocytes (Figure 4A,B,D). Interestingly, the activation of AMPK and its downstream target ACC was significantly downregulated, consistent with their negative correlation with PPARγ and C/EBP expression during adipogenesis [51,52]. However, SPSL treatment dose-dependently reversed these patterns, impacting the AMPK pathway as well as PPARγ and its downstream targets for lipogenesis, such as SREBP1 and ATGL (Figure 4A–G). Previous studies have demonstrated that fasting or agonist-induced AMPK activation reduces PPARγ transcription in adipose tissues in vivo [53]. This SPSL-induced anti-adipogenic effect was reversed by AMPK inhibition via Compound C co-incubation during adipocyte differentiation (Figure 5A,B). Therefore, SPSL-stimulated AMPK activation in adipocytes likely regulates adipogenesis and lipogenesis.

Intracellular energy homeostasis through catabolic pathways that balance ATP levels, primarily through enhanced mitochondrial respiration and biogenesis [35]. AMPK regulates mitochondrial ATP production by modulating NAD^+^ metabolism in metabolic organs, including adipose tissues [52]. To investigate the effects of SPSL on energy metabolism in vivo, we selected a zebrafish model [31]. SPSL application led to dose-dependent increases in energy expenditure through AMPK activation-linked NAD^+^ changes, as evidenced by co-incubation with Alamar Blue, a colorimetric indicator of NAD^+^ production (Figure 6A,B). SPSL also depicted a remarkable anti-hyperlipidemia effect with reduced lipid accumulation in zebrafish (Figure 6D,F). Although the specific mechanism of seaweed sulfated polysaccharide-induced energy expenditure and weight loss remain unclear, *Sargassum fusiforme* fucoidan has been shown to inhibit fat accumulation via UCP1-linked energy expenditure enhancement [54]. While our research did not provide evidence that SPSL activation of UCP1, the reported co-activation of AMPK and UCP1 through the PGC1α pathway suggests further investigation into the role of UCP1 in the anti-obesity effects of SPSL [55].

In conclusion, our findings demonstrate that SPSL reduces fat accumulation via the AMPK-PPARγ pathway and enhances energy expenditure. However, further research is needed to elucidate the specific monosaccharide linking exact structures with sulfate group locations of SPSL. The composition of monosaccharides in fucoidan is known to vary significantly depending on the seaweed source, which can influence its bioactivity [15]. Thus, future investigations will focus on applying our established extraction and analysis methods to sulfated polysaccharides from various brown algae species and explore the correlation between specific monosaccharide and glycan structures and their associated anti-obesity effects. Comprehensive structural analysis of these polysaccharides could unveil valuable insights, potentially leading to the development of algal sulfated polysaccharides as a reliable source of HFFs with therapeutic potential in metabolic disorders.

## 4. Materials and Methods

### 4.1. Chemicals and Reagents

Dulbecco’s modified Eagle’s medium (DMEM), bovine serum, fetal bovine serum (FBS), and antibiotics (Anti-Anti 100×) were purchased from Gibco (Carlsbad, CA, USA). Primary and secondary antibodies were purchased from Cell Signaling Technology (Danvers, MA, USA) and Santa Cruz Biotechnology (Santa Cruz, CA, USA), respectively. Dexamethasone, isobutyl-1-methylxanthine (IBMX), insulin, oil red O (ORO), 37% formaldehyde solution, Compound C, Alamar Blue, RIPA buffer, egg-yolk powder, and bovine serum albumin (BSA) were obtained from Sigma-Aldrich (St. Louis, MO, USA). Food-grade Celluclast (1.5 L) was purchased from Novozyme Nordisk (Bagsvaerd, Denmark). SL was botanized at Seongsan Ilchulbong in Jeju province of South Korea (lat. 33°27′10.4″, long. 126°56′34.0″ E) and the collected SL was dried using hot-air drying after three times rinsing by tap-water for desalination. All other chemicals used in this study were purchased from Daejung Chemicals and Metals (Siheung-si, Republic of Korea).

### 4.2. Preparation of Celluclast-Assisted SL Hydrolysate and Sulfated Polysaccharides from SLC

Celluclast-assisted hydrolysis was carried out using SL to prepare SLC; 10 g powder of wind-blow-dried SL was agitated in distilled water at 1% volume and adjusted pH 4.5 to prepare the Celluclast working concentration (0.01%, 50 °C, 24 h). After 24 h, SLC was centrifuged (5000× *g*, 4 °C, 10 min) and filtered, and the supernatant was heat inactivated (100 °C, 10 min) and neutralized. Subsequently, SLC was precipitated by adding 95% ethanol for homogenization (200% of volume, 4 °C, 12 h) and centrifuged (5000× *g*, 4 °C, 10 min) to obtain SPSL following a previously described method with minor changes [56]. The ethanol precipitate containing SPSL was evaporated and freeze-dried for further use.

### 4.3. Proximate Composition Analysis

Approximate polyphenol, polysaccharide, protein, and sulfate compositions were determined using the analytical procedure of the Association of Official Analytical Chemists. The polysaccharide value was assessed using a phenol-sulfuric acid colorimetric assay [57]. The total phenol content was evaluated using the Folin—Ciocalteu analysis [58]. The total protein concentration was analyzed using a Pierce BCA protein analysis kit. The sulfate contents were estimated using the barium chloride-gelatin reagent method to determine the turbidimetry of the specimens [59].

### 4.4. Monosaccharide Composition Analysis Protocol

Trifluoroacetic acid solution (2 M) was used for acid-hydrolysis of all samples at 120 °C. Next, the treated samples were cooled to room temperature and filtered immediately after neutralization. Subsequently, the samples were injected into a Dionex Bio-LC system (Thermo Fisher Scientific, Waltham, MA, USA) linked to a Dionex ED 50 electrochemical detector (Thermo Fisher Scientific). Isocratic elution (18 mM NaCl) was performed using a CarboPac PA1 column (4.5 × 50 mm) at a flow rate of 1 mL/min. A monosaccharide standard mixture (L-fucose, L-rhamnose, L-arabinose, L-galactose, D-glucose, L-xylose, and D-fructose) was used as the calibration agent to compare the retention times of the samples (Figure 2).

### 4.5. Estimation of Molecular Weight Distribution of Sulfated Polysaccharide

All samples with the sulfate polysaccharide standards to compare the molecular weight were loaded in 5% agarose gel in 1x TBE buffer (10.8 g Tris, 5.5 g Boric acid, 4 mL 0.5 M Na_2_EDTA/1L of ddiH_2_O) to determine the molecular weight differences of anionic sulfated polysaccharide existed in sample. The electrophoresis was performed under the condition (100 V, 0.5 to 1 h). After running in the agarose gel, the gel was stained in 0.02% o-Toluidine in 3% acetic acid for 12 h. Next the blue stained gel was soaked in 3% acetic acid in ddiH_2_O destaining buffer by repeated washing on the rocker. And the bands indicating polysaccharide was scanned for visualization. MW: 50-500 kDa (Dextran sulfate, D8906, Sigma), MW: 60 kDa (Chondroitin 6-sulfate, D4384, Sigma), MW: 8 kDa (Dextran sulfate, D4911, Sigma).

### 4.6. Fourier-Transform Infrared Spectroscopy (FT-IR) Analysis

FT-IR spectroscopy, using a Bruker FTIR Alpha II spectrometer (Bruker, Germany), was employed to characterize the sulfated polysaccharides extracted from edible brown algae. Prior to analysis, samples were thoroughly dried, with the option of deuterated water exchange to minimize interference from water absorption. Spectra were acquired within the range of 500–2000 cm^−1^ with a resolution of 4 cm^−1^, averaging 32–64 scans per sample. Data analysis focused on the identification of characteristic absorption bands associated with sulfate groups (S=O stretching at 1220–1270 cm^−1^; C-O-S stretching at 800–850 cm^−1^) as well as C-O stretching (1600–1650 cm^−1^) and the fingerprint region (below 1000 cm^−1^). Obtained spectra were compared to reference spectra of known sulfated polysaccharides to aid in structural characterization. Multiple replicates were analyzed to ensure reproducibility.

### 4.7. Identification of Sulfated Glycans in SPSL Using UPLC-Q-TOF MS/MS Based Fucoidan Library

The preparation for the identification of sulfated glycans in SPSL using MS spectrometry based on a fucoidan library followed our previous research [60]. Briefly, SPSL underwent acid hydrolysis in 0.01 M HCl at 60 °C for 18 h and was then neutralized with 0.01 M NaOH. The acidic hydrolyzed SPSL was purified using a Supelclean™ ENVI-Carb™ (Bellefonte, PA, USA) SPE tube and filtered through a 0.45 µm membrane and diluted to 1 mg/mL for further analysis. UPLC analysis was performed using an ACQUITY^®^ UPLC BEH C18 column (2.1 × 100 mm, 1.7 µm) (Waters Corporation, Wilmslow, UK). The mobile phases used were A: ddH_2_O with 0.1% formic acid and B: acetonitrile with 0.1% formic acid. The injection volume was set at 5 µL, and the flow rate was 0.2 mL/min. The gradient condition of UPLC was the following conditions: 100% A by 3 min, 75% A by 15 min, 50% A by 18 min, 0% A by 23 min, and 100% A by 23.1–25 min.

ESI conditions were conducted in negative mode with a capillary voltage of 4.5 kV, a charging voltage of 2 kV, a desolvation temperature of 200 °C, and a nitrogen gas flow rate of 8.0 L/min. UPLC-Q-TOF MS/MS analysis was carried out in the *m*/*z* 50–1600 range using a MassLynx v4.1 SCN888 (Waters Corp.). Data analysis was performed using UNIFI^®^ v1.8.0 (Waters Corp.). Subsequently, the information of *m*/*z* indicating major glycan construction in SPSL was applied to the fucoidan library to predict the form of sulfated glycan structures with reference to the previous study [60]. The abbreviations of glycans were presented to L-fucose (Fuc), hexose (Hex), and L-xylose (Xyl).

### 4.8. Cell Culture and Determination of Adipocyte Differentiation

The murine preadipocyte cell line 3T3-L1 was purchased from the American Type Culture Collection (ATCC, Manassas, VA, USA). The 3T3-L1 cells were incubated in DMEM containing 10% bovine serum and 1% antibiotics at 37 °C at 5% CO_2_ in humid conditions. Cells were differentiated under the described condition briefly, and 3T3-L1 pre-adipocytes were seeded to 1 × 10^5^ cells/mL and maintained for 72 h. Thereafter, the medium was replaced with a differentiation medium (MDI; DMEM containing 10% FBS, 1% anti-anti, 0.5 mM of IBMX, 1 μM of dexamethasone, and 850 nM of insulin) for 48 h. Subsequently, the cell culture was changed to 850 nM of insulin-containing medium once every 48 h. The rate of adipogenesis was determined via ORO staining. First, fully differentiated 3T3-L1 adipocytes were fixed with 10% formalin for 1 h, and every well was rinsed with 60% 2-propanol in ddiH_2_O. Wells were stained using 0.6% ORO in 2-propanol for 30 min at room temperature and washed with ddiH_2_O three times. Then, the ORO-stained lipids were lysed using absolute 2-propanol, and the absorbance was detected at 540 nm using a Synergy HTX Multi-Detection Microplate Reader (Agilent Technologies, Palo Alto, CA, USA), followed by observation under an optical microscope. Every experiment was performed on the seventh day after initiating differentiation.

### 4.9. Immunoblotting Assay

A RIPA buffer was used for cell lysis, and protein levels were quantified using SDS-PAGE [61]. Cultured cells were lysed in radioimmunoprecipitation assay (RIPA) lysis buffer containing a protease and phosphatase inhibitor cocktail. The protein concentration was evaluated using the BCA assay and heated at 95 °C for 5 min. The proteins were separated using 10% SDS acrylamide gels, and the separated proteins were transferred onto polyvinylidene fluoride (PVDF) membranes. The membrane was blocked with EveryBlot Blocking buffer at room temperature for 15 min. The blocked membrane was incubated with primary antibodies (PPARγ; CST#2435s, C/EBPα; CST#2295s, SREBP-1; sc-365513, ATGL; CST#2439s, p-AMPKα; CST#2531s, AMPKα; CST#2603s, p-ACC; CST#3661s, ACC; CST#3676s, and β-actin; sc-47778) at 4 °C for overnight. The membrane was then rinsed with TBST and incubated with secondary antibodies (Anti-mouse IgG, HRP-linked antibody; 7076s and Anti-rabbit IgG, HRP-linked antibody; 7074s) at room temperature for 1 h. Protein expression was visualized using Fusion FX (Vilber Lourmat, Marne-la-Vallée, France) at Korea Basic Science Instititue (KBSI) Gwangju center.

### 4.10. Zebrafish Maintenance

Mature zebrafish (one-year-old) were purchased from a commercial breeder (Jeju Aquarium, Jeju, Republic of Korea). Mature wild-type zebrafish were placed in a 5 L cleared tank at 28.5 ± 1 °C and 14-:10-h dark/light condition. Zebrafish were fed twice per day with TetraBits complete diet (D-49304, Melle, Germany).

### 4.11. Energy Expenditure Assay Using Zebrafish Embryo

On the day before breeding, one female and two males were grouped with a membrane between them. The following day, ovulation was induced by light stimulation, and only those obtained within 30 min were used in the experiment. The embryos were then transferred to a Petri dish containing the E3 embryo medium (290 mg NaCl, 13 mg KCl, 5.8 mg CaCl_2_°2H_2_O, and 82 mg MgCl_2_°6H_2_O in 1 L ddiH_2_O, pH 7.2). Two days post-fertilization (dpf), zebrafish embryos were rinsed three times with the sterile filtered E3 embryo medium. All viable zebrafish embryos were placed in beakers. Single zebrafish embryos were picked using a fine-tip disposable transfer pipette and gently transferred to each well of a clear-bottom 96-well plate. Two types of blanks containing the embryos were prepared. Then, 300 µL of carrier medium was discarded from the wells, and 300 µL of 1% Alamar Blue was added back to the E3 embryo medium, including the blank well. Next, 3 µL of 100× sample solution was added into the desired wells, the vehicle was added to blank wells, and they were incubated for 48–72 h in a humidified 28.5 °C dark chamber to prevent photobleaching. After the indicated time, 100 µL of supernatant was collected into clear-bottomed 96-well plates to determine the fluorescence value at 530/590 nm (Ex/Em), and the detected fluorescence intensity was calculated using the following equation:$$Energy\;expenditure\;(Fold\;change)=\frac{Sample–treated\;wells-Empty\;blank\;wells}{Vehicle–treated\;blank\;wells-Empty\;blank\;wells}$$

### 4.12. High Cholesterol Diet Assay Using Zebrafish Embryo

The 5 dpf zebrafish larvae were used to investigate the anti-lipid accumulation effect of SPSL. The experimental protocol was derived from previous research [62]. In brief, 100 mL of 0.1% cholesterol, with or without the test specimens, in the E3 embryo medium was distributed into individual beakers based on the designated groups. Thirty zebrafish larvae were placed in each group of beakers, and they were incubated for 48 h at 28 °C, following a day/night cycle. After 48 h, the zebrafish were briefly immersed in 0.003% MS-222 (Sigma-Aldrich) for three min to anesthetize them. The anesthetized zebrafish were then fixed in 4% neutral buffered formalin overnight in a cold room. On the following day, the fixed zebrafish were stained for 15 min in an ORO working solution, which consisted of 300 µL of 0.5% ORO in 100% 2-propanol, along with an additional 200 µL of ddiH_2_O. Subsequently, the stained zebrafish were rinsed three times in a solution of 0.2% Tween-20 in 1× PBS for five min each. Finally, images of the zebrafish were acquired using an automated live-cell imager (Lion Heart, Agilent technologies) while they were submerged in chilled 100% Glycerin. The level of lipid accumulation was determined by incubating the oil red O-stained zebrafish larvae in a well with 100% 2-propanol. After a 10-min incubation period, the supernatant was collected and transferred to transparent-bottomed 96-well plates for the measurement of light absorbance at 450 nm using a microplate reader (Synergy HTX, Agilent Technologies). The triglyceride level of homogenized zebrafish larvae in 1× PBS was determined according to the manufacturer protocol for the specified assessment kit (AM157S; Asan Pharm. Co., Ltd., Seoul, Republic of Korea).

### 4.13. Statistical Analysis

The statistical significance of observed differences was assessed through rigorous hypothesis testing. A one-way analysis of variance (ANOVA) was employed for multiple group comparisons, while unpaired *t*-tests were utilized for pairwise comparisons (e.g., Figure 2B sulfate content). All statistical analyses were conducted using GraphPad Prism 10 software (GraphPad Software Inc., CA, USA). All data are presented as means ± SEM (*^,#^
*p* < 0.05; **^,##^
*p* < 0.01; ***^,###^
*p* < 0.001; and ****^,####^
*p* < 0.0001) (*n* = 3).

## 5. Conclusions

This study aimed to highlight the industrial potential of SPSL obtained from an edible brown alga, which is abundant worldwide but limited in use due to insufficient research. SPSL exhibited distinctive characteristics, including high polysaccharide content, abundance of L-fucose, and elevated sulfate content. SPSL demonstrated a potent anti-lipogenesis effect in 3T3-L1 adipocytes, inhibiting key regulators of adipogenesis. We observed SPSL-reduced adipogenesis in adipocytes, along with regulation of AMPK activation and PPARγ suppression. In addition, SPSL supplementation in zebrafish upregulated AMPK activation-linked NAD^+^ changes, a significant marker of energy expenditure improvement and reduced lipid deposition, a marker of the anti-hyperlipidemia effect. Taken together, these results strongly support the scientific evidence for the development of SPSL as an industrial application for metabolic stress relief.

## Figures and Tables

**Figure 1 ijms-25-09738-f001:**
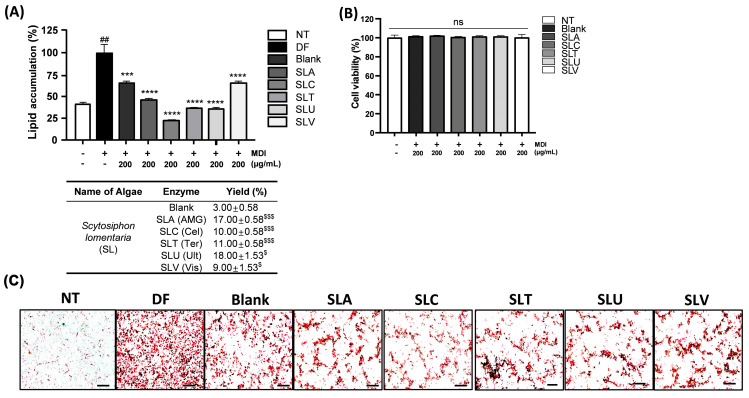
Anti-adipogenic effect of the polysaccharide-rich SL hydrolysates using five different enzymes in 3T3-L1 cells. (**A**) Lipid accumulation percentage in ORO staining in 3T3-L1 cells treated with vehicle and five different samples during adipocyte differentiation and enzymatic hydrolysate yields. NT: untreated; DF: differentiated control; SL: *S. lomentaria*; Blank: No enzyme-assisted SL hydrolysis; SLA: AMG-assisted SL hydrolysate; SLC: Celluclast-assisted SL hydrolysate; SLT: Termamyl-assisted SL hydrolysate; SLU: Ultraflo-assisted SL hydrolysate; SLV: Viscozyme-assisted SL hydrolysate (*n* = 4). (**B**) Adipocyte differentiation rate of each hydrolysate in 3T3-L1 cells under ORO staining. (**C**) Cell viability of each hydrolysate during adipocyte differentiation (*n* = 3). Scale bar: 50 μm. ns; not significant. MDI: IBMX, dexamethasone, insulin cocktail. *** *p* < 0.001 and **** *p* < 0.0001 vs. DF, ^##^
*p* < 0.01 vs. NT, ^$^
*p* < 0.05 and ^$$$^
*p* < 0.001 vs. Blank. All data are presented as means ± SD.

**Figure 2 ijms-25-09738-f002:**
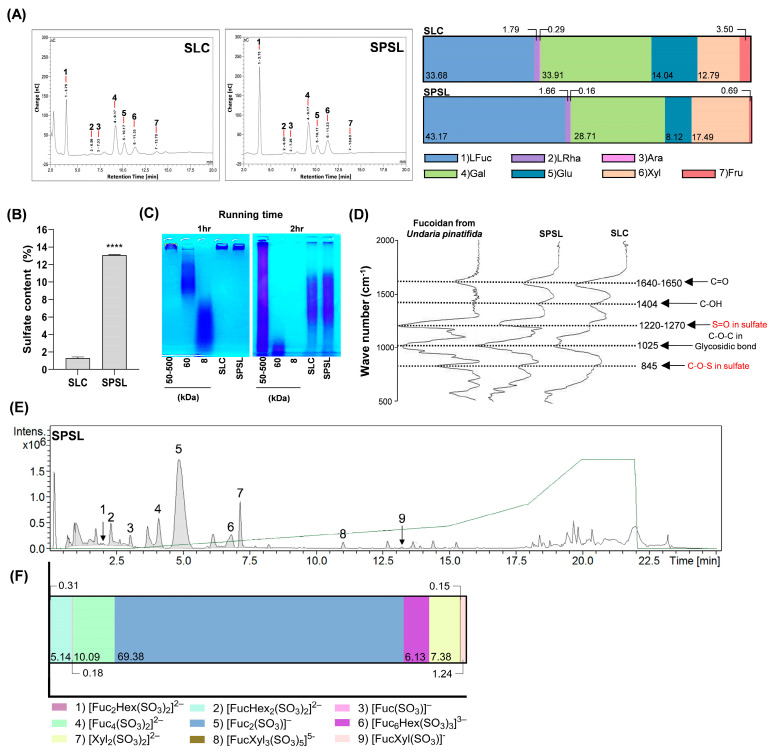
The characterization of sulfated polysaccharides from SLC (SPSL). (**A**) The percentages of proximate composition rates and neutral monosaccharide compositions were evaluated in SLC and SPSL, respectively. Fuc: L-Fucose, Rha: L-Rhamnose, Ara: L-Arabinose, Gal: L-Galactose, Glu: D-Glucose, Xyl: L-Xylose, Fru: D-Fructose. (**B**) The percentage of sulfate content in SLC and SPSL. Data are indicated as means ± SD (*n* = 3). (**C**) Molecular weight analysis of SPSL. The molecular weight distribution of the SPSL along with SLC was assessed by agarose gel electrophoresis with three different standards known as its molecular weights. MW: 50−500 kDa (Dextran sulfate, D8906, Sigma-Aldrich, MO, USA). MW: 60 kDa (Chondroitin 6-sulfate, D4384, Sigma-Aldrich). MW: 8 kDa (Dextran sulfate, D4911, Sigma-Aldrich). (**D**) The comparative alignments of FT-IR spectrum from SLC, SPSL, and commercial fucoidan. (**E**) The chromatogram of SPSL acid hydrolysate for glycan analysis. (**F**) The percentages of major sulfated glycan constituents in SPSL. **** *p* < 0.0001 vs. SLC.

**Figure 3 ijms-25-09738-f003:**
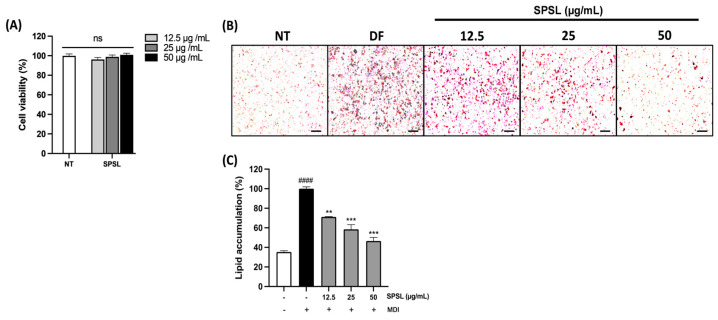
The anti-adipogenesis effect of SPSL in 3T3-L1 adipocytes. (**A**) The percentage of cell viability of the series of doses of SPSL-treated 3T3-L1 adipocytes. (**B**) The visualization of three different doses of SPSL exposed 3T3-L1 cells by ORO staining. (**C**) The quantification of lipid accumulation rate by ORO staining in the three different doses of SPSL-exposed 3T3-L1 cells. Scale bar: 50 μm. ns; not significant. MDI: IBMX, dexamethasone, insulin cocktail. ** *p* < 0.01 and *** *p* < 0.001 vs. DF; ^####^
*p* < 0.0001 vs. NT. All data are expressed as means ± SD (*n* = 3).

**Figure 4 ijms-25-09738-f004:**
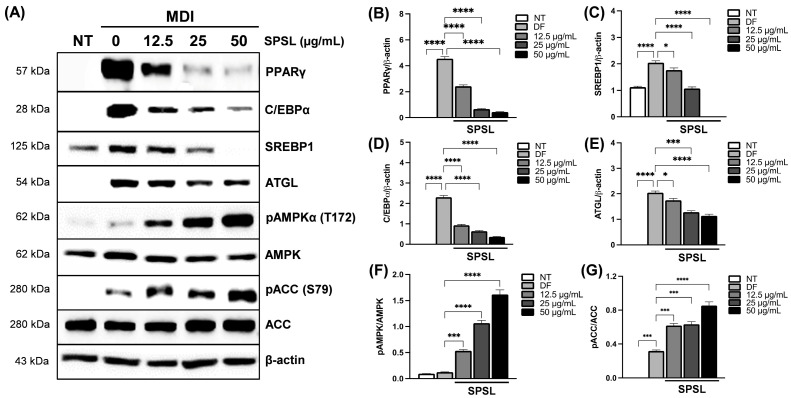
The adipogenesis modulators were controlled by SPSL treatment in 3T3-L1 adipocytes. (**A**) The immunoblotting for proteins related to adipogenesis modulation controlled by SPSL treatment. (**B**) The quantification of changes in PPARγ expressions. (**C**) The quantification of changes in SREBP-1 expression. (**D**) The quantification of changes in C/EBPα expression. (**E**) The quantification of changes in ATGL expression. (**F**) The quantification of changes in pAMPKα expression. (**G**) The quantification of changes in pACC expression. MDI: IBMX, dexamethasone, insulin cocktail. * *p* < 0.05, *** *p* < 0.001, and **** *p* < 0.0001. All data are presented as means ± SD (*n* = 2).

**Figure 5 ijms-25-09738-f005:**
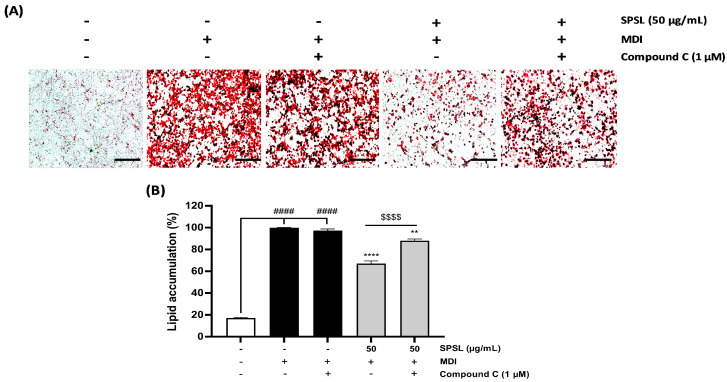
AMPK inhibition decreased the anti-adipogenesis effect of SPSL in adipocytes. (**A**) The visualization of fully- differentiated 3T3-L1 adipocytes that were incubated SPSL and SPSL + Compound C, respectively. (**B**) The quantification results of the rate of lipid accumulation in 3T3-L1 adipocytes that were incubated SPSL and SPSL with Compound C, respectively. Scale bar: 100 μm. MDI: IBMX, dexamethasone, insulin cocktail. ** *p* < 0.01 and **** *p* < 0.0001 vs. DF; ^####^
*p* < 0.0001 vs. NT, ^$$$$^
*p* < 0.0001 vs. SPSL (±Compound C). All data are presented as means ± SD (*n* = 3).

**Figure 6 ijms-25-09738-f006:**
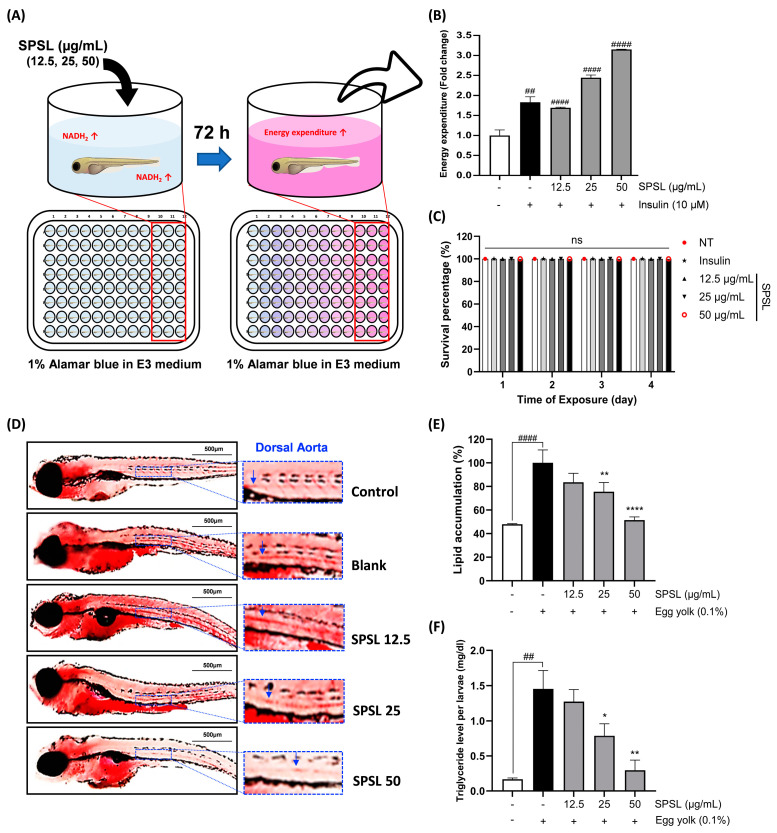
Improvement of energy expenditure rate and lipid accumulation in zebrafish by SPSL treatment. (**A**) The assay for energy expenditure assessment using Alamar Blue staining in a zebrafish model. (**B**) Energy expenditure levels after 72 h of SPSL treatment. (**C**) The survival rate of zebrafish exposed to different SPSL concentrations for 72 h. ns; not significant. (**D**) Oil red O-stained zebrafish were supplemented with an egg yolk along with SPSL in a dose-dependent manner. The blue arrow within blue dash lined box indicates a dorsal aorta portion in zebrafish with 10× magnification. (**E**) The level of lipid accumulation in zebrafish supplemented an egg yolk for 48 h. (**F**) The triglyceride level in zebrafish larvae supplemented an egg yolk for 48 h. Scale bar: 500 μm. ^##^
*p* < 0.01 and ^####^
*p* < 0.0001 vs. NT; * *p* < 0.05, ** *p* < 0.01, and **** *p* < 0.0001 vs. Blank. All data are presented as means ± SEM (*n* = 3).

**Table 1 ijms-25-09738-t001:** The proximate composition of SLC and SPSL. The proximate compositions (polysaccharides, proteins, and polyphenols) of SLC and SPSL are presented in the table. All data are presented as means ± SD (*n* = 3). * *p* < 0.05 and ** *p* < 0.01 vs. SLC.

	Proximate Composition (%)		
Sample	Polysaccharides	Proteins	Polyphenols
SLC	12.49 ± 0.26	9.89 ± 0.38	0.85 ± 0.04
SPSL	41.99 ± 0.65 **	12.10 ± 1.65 *	0.48 ± 0.04

**Table 2 ijms-25-09738-t002:** Glycan structures identified in SPSL using glycan library prediction. The predicted glycan structures in SPSL, corresponding to their retention times in the chromatogram (Figure 2E), are presented in the table below.

No.	Predicted Glycan Structure	Area	*m*/*z*	Exact Mass (g·mol^−1^)
1	[Fuc_2_Hex(SO_3_)_2_]^2−^	174,865	315.0391	630.0783
2	[FucHex_2_(SO_3_)_2_]^2−^	2,863,978	323.0366	646.0732
3	[Fuc(SO_3_)]^−^	97,920	243.018	243.018
4	[Fuc_4_(SO_3_)_2_]^2−^	5,618,608	380.0706	760.1413
5	[Fuc_2_(SO_3_)]^−^	38,633,162	389.0759	389.0759
6	[Fuc_6_Hex(SO_3_)_3_]^3−^	3,410,449	431.0865	1293.2595
7	[Xyl_2_(SO_3_)_2_]^2−^	4,109,871	219.9971	439.9942
8	[FucXyl_3_(SO_3_)_5_]^5−^	83,677	190.9886	954.9429
9	[FucXyl(SO_3_)]^−^	691,885	375.0602	375.0603

## Data Availability

The original data presented in the study are openly available in https://jejuunivackr-my.sharepoint.com/:f:/g/personal/localman_office_jejunu_ac_kr/EnfMBJktRp5Jn-aiCl4x8GsBtIyem6FLsK_erwtPZ9pVWQ?e=hEQT9p (accessed on 8 August 2024).

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
