# Peer review of "L-Fucose-Rich Sulfated Glycans from Edible Brown Seaweed: A Promising Functional Food for Obesity and Energy Expenditure Improvement"

_ijms, 2024, doi:10.3390/ijms25179738_

Round 1

Reviewer 1 Report

Comments and Suggestions for Authors

Dear the Editor

Hyun J et al reported the anti-obesity effect of SPSL, a sulfated polysaccharide derived from the brown seaweed Scytosiphon lomentaria. The role of SPSL in vitro was demonstrated in adipocytes (Fig. 1,3-5): SPSL downregulates AMPK and PPARgamma-mediated signaling pathway, leading to an attenuated differentiation/groth of adipocytes. In vivo, these authors found an increase in energy expenditure and lipid accumulation in zebrafish (Fig. 6). The results of chemical characterization was summarized in Fig. 2. Overall, this manuscript was well-organized with sufficient description of Method section. In particular, the number of sulfates was reasonablly speculated using IR data. Also, in vitro study has been performed with reasonable scientific basis. The limitation of this study could be involved in the partial identification of materials: perharps, the additional information will be published elsewhere.

Major concerns:

1) In 4.1, the source of starting material (ie Scytosiphon lomentaria) needs to be described. Is this purchased at the local supermarket or is this harvested at specific area in author's country?

2) In Fig 2(E), please described the method of detection for chromatograpy.

3) Please provide some speculative comments on sulfate isomers.

4) Please add some discussion on health functional foon (HFF) on SPSL. This is found in the Introduction, but not in the Discussion section.

Minor concerns:

1) In L426, there are some typos.

Author Response

Responses to Technical Check Results

Journal: International Journal of Molecular Sciences (ISSN 1422-0067)

REF: ijms-3186639

Title: L-Fucose-Rich Sulfated Glycans from Edible Brown Seaweed: A promising Functional Food for Obesity and Energy Expenditure Improvement

Hyun J et al reported the anti-obesity effect of SPSL, a sulfated polysaccharide derived from the brown seaweed Scytosiphon lomentaria. The role of SPSL in vitro was demonstrated in adipocytes (Fig. 1,3-5): SPSL downregulates AMPK and PPARgamma-mediated signaling pathway, leading to an attenuated differentiation/groth of adipocytes. In vivo, these authors found an increase in energy expenditure and lipid accumulation in zebrafish (Fig. 6). The results of chemical characterization was summarized in Fig. 2. Overall, this manuscript was well-organized with sufficient description of Method section. In particular, the number of sulfates was reasonablly speculated using IR data. Also, in vitro study has been performed with reasonable scientific basis. The limitation of this study could be involved in the partial identification of materials: perharps, the additional information will be published elsewhere.

Comment 1:

Major concerns:

1) In 4.1, the source of starting material (ie Scytosiphon lomentaria) needs to be described. Is this purchased at the local supermarket or is this harvested at specific area in author's country?

Response:

We appreciate your valuable insights. We added the material information of Scytosiphon lomentaria in line 385-387.

Line 385-387 “SL was botanized at Seongsan Ilchulbong in Jeju province of South Korea (lat.33°27'10.4", long.126°56'34.0"E) and the collected SL was dried using hot-air drying after three times rinsing by tap-water for desalination.”

Comment 2:

2) In Fig 2(E), please described the method of detection for chromatography.

Response:

Thank you for your dedicate comment and suggestion regarding the chromatography section. In line 438-442, we inputted the detailed information of our chromatography condition.

Line 438-442 “The mobile phases used were A: ddH2O with 0.1% formic acid and B: acetonitrile with 0.1% formic acid. The injection volume was set at 5 µL, and the flow rate was 0.2 mL/min. The gradient condition of UPLC was following conditions: 100% A by 3 min, 75% A by 15 min, 50% A by 18 min, 0% A by 23 min, and 100% A by 23.1-25 min.”

Comment 3:

3) Please provide some speculative comments on sulfate isomers.

Response:

We are deeply appreciated to your thoughtful consideration onto key points of our manuscript. The sulfate isomers in the context of sulfated polysaccharides which are mainly constituted fucoidan refer to the diversity in the positions where sulfate groups attach to fucose monosaccharide. Fucose, the main component of fucoidan, can have sulfate groups attached at various positions. This variation in sulfate group positions creates structural diversity in fucoidan, which can influence its biological activities and functions. For example, sulfate groups can bind to the 2nd, 3rd, or 4th carbon of fucose, and the combination of these sulfate group positions generates various sulfate isomers. Each isomer may exhibit different potencies for specific biological activities, such as antioxidant, anti-inflammatory, and anticancer effects. Therefore, analyzing and understanding the sulfate isomers of fucoidan can reveal the structure-activity relationship of fucoidan and provide valuable information for the development of functional foods or pharmaceuticals utilizing fucoidan.

While the precise positional elucidation of sulfate isomers within SPSL remains elusive, the identification of sulfate groups associated with monosaccharides and the fundamental basis structural unit establishes a foundation for future investigations. Through subsequent research endeavors, we anticipate the definitive characterization of these isomers, facilitating the procurement of marine-derived food ingredients possessing a well-defined structure-function relationship. Also, this context is reflected in line 317-328 of our manuscript.

Line 317-328 “Sulfate groups in sulfated polysaccharides, primarily composed of fucoidan, refer to the diverse positions where these groups attach to algal monosaccharides. Fucose, the main monosaccharide in fucoidan, can have sulfate groups attached at various positions, such as the 2nd, 3rd, or 4th carbon [1,2]. This diversity in sulfate group attachment sites creates structural variations in fucoidan, potentially influencing its biological functions, including antioxidant, anti-inflammatory, and anti-cancer effects [3]. Numerous studies have revealed the relationship between fucoidan's structure and specific physiological activities [2,4,5]. While the precise positional elucidation of sulfate isomers within SPSL remains challenging, identifying sulfate groups associated with specific monosaccharides and the fundamental structural unit in SPSL, which exhibits remarkable anti-obesity effects, lays a foundation for future investigations.”

Comment 4:

4) Please add some discussion on health functional food (HFF) on SPSL. This is found in the Introduction, but not in the Discussion section.

 Response:

Thank you for your valuable feedback. I believe your insights will significantly improve our readability for readers. In Line 284 to 287, we slightly modified the text to emphasize our research purpose regarding development for marine-derived HFF ingredient.

Line 284-287 “Consequently, public interest in natural product-derived HFFs with fewer side effects and clear improvements, compared to conventional drugs, has fueled interest in exploring the potential of marine-derived biomolecules.”

Comment 5:

Minor concerns:

1) In L426, there are some typos.

Response:

Thank you for your careful comments. Your input will help us provide more accurate information to our readers. We edited follow your comments.

Line 454-455 “The 3T3-L1 cells were incubated in DMEM containing 10% bovine serum and 1% antibiotics at 37 °C at 5% CO2 at humid conditions.”

References

  1. Li, B.; Lu, F.; Wei, X.; Zhao, R. Fucoidan: structure and bioactivity. Molecules (Basel, Switzerland) 2008, 13, 1671-1695, doi:10.3390/molecules13081671.
  2. Ale, M.T.; Meyer, A.S. Fucoidans from brown seaweeds: an update on structures, extraction techniques and use of enzymes as tools for structural elucidation. RSC Advances 2013, 3, 8131-8141, doi:10.1039/C3RA23373A.
  3. Shi, L. Bioactivities, isolation and purification methods of polysaccharides from natural products: A review. Int J Biol Macromol 2016, 92, 37-48, doi:10.1016/j.ijbiomac.2016.06.100.
  4. Lee, J.H.; Park, J.E.; Han, J.S. Fucoidan Stimulates Glucose Uptake via the PI3K/AMPK Pathway and Increases Insulin Sensitivity in 3T3-L1 Adipocytes. Journal of Life Science 2021, 31, 1-9.
  5. Liu, Y.; Wang, N.; Tian, Y.; Chang, Y.; Wang, J. Fucoidans from Thelenota ananas with 182.4 kDa Exhibited Optimal Anti-Adipogenic Activities by Modulating the Wnt/β-Catenin Pathway. Journal of Ocean University of China 2021, 20, 921-930, doi:10.1007/s11802-021-4681-8.

Reviewer 2 Report

Comments and Suggestions for Authors

This paper describes a study exploring the anti-obesity effects of sulfated polysaccharides (SPSLs) from brown algae. The worldwide increase in obesity, exacerbated by the lack of exercise associated with the COVID-19 pandemic, has created a need for safer and more effective treatments. Studies have shown that SPSL, extracted from Scytosiphon lomentaria (SL), has the ability to promote energy expenditure and decrease fat accumulation. In particular, SPSL inhibited lipogenesis via the AMPK-PPARγ pathway, and reduced lipid accumulation and increased energy expenditure in a study using zebrafish. This makes SPSL a potentially promising functional food material for the treatment of obesity and improvement of metabolic disorders. However, further mechanism elucidation and preclinical studies are needed, and its industrial application is expected in future research. This manuscript presents an interesting perspective, but there are several points that require improvement. My feedback is intended to be constructive and beneficial for enhancing this study.

Major feedback:

1. Inhibition of drugs related to adipocyte differentiation and lipid accumulation by SPSLs.

I have one concern about the inhibition of drugs related to adipocyte differentiation and lipid accumulation by SPSLs. I think that SPSL is like dietary fiber and adsorbs other components such as triglycerides, cholesterol, and bile acids. Therefore, could it be that SPSL did not reduce lipid accumulation in adipocytes and zebrafish, but rather attenuated the effects of drugs that promote fat accumulation? Please discuss this point in the Discussion or Limitation of this study.

2. SPSLs is high-molecular weights

The author aims to develop SPSL as a health functional food. If so, the cell experiments in this study seem to be far from the purpose, since SPSL is of high molecular weight and cannot be absorbed at this size. If the authors are committed to this goal, they should describe the limitations of this cell study.

3. Abbreviations

AMPK and PPAR in Abstract are being used without the presentation of official names. I do not know the rules of this journal, but if this is the normal rule, the official name and abbreviation should be presented first, and then the abbreviation should be used thereafter.

4. Title

The “L” of L-fucose is small capital.

5. Fig. 6C

Points other than NT are hidden and not visible. Therefore, I think this graph is inappropriate. The author should consider changing this graph to a bar chart.

6. Statistical analysis

The authors used the T-test as comparison between multiple groups. But, the t-test is a test method used for comparisons between two groups. Thus, please change to another test. T-test is very prone to significant differences, so using the correct method should result in no significant differences in many items. Therefore, it is necessary to use the correct statistical analysis. If the statistical processing method changes, the conclusion will also change.

Author Response

Responses to Technical Check Results

Journal: International Journal of Molecular Sciences (ISSN 1422-0067)

REF: ijms-3186639

Title: L-Fucose-Rich Sulfated Glycans from Edible Brown Seaweed: A promising Functional Food for Obesity and Energy Expenditure Improvement

Dear Editor,

Thank you for your useful comments and suggestions regarding our manuscript. We have corrected the manuscript accordingly.

The revised contents are highlighted in red in the revised manuscript. All authors have read and approved the revised version of the manuscript. We appreciate your consideration for publishing our manuscript in International Journal of Molecular Sciences.

Reviewer 1

This paper describes a study exploring the anti-obesity effects of sulfated polysaccharides (SPSLs) from brown algae. The worldwide increase in obesity, exacerbated by the lack of exercise associated with the COVID-19 pandemic, has created a need for safer and more effective treatments. Studies have shown that SPSL, extracted from Scytosiphon lomentaria (SL), has the ability to promote energy expenditure and decrease fat accumulation. In particular, SPSL inhibited lipogenesis via the AMPK-PPARγ pathway, and reduced lipid accumulation and increased energy expenditure in a study using zebrafish. This makes SPSL a potentially promising functional food material for the treatment of obesity and improvement of metabolic disorders. However, further mechanism elucidation and preclinical studies are needed, and its industrial application is expected in future research. This manuscript presents an interesting perspective, but there are several points that require improvement. My feedback is intended to be constructive and beneficial for enhancing this study.

Comment 1:

  1. Inhibition of drugs related to adipocyte differentiation and lipid accumulation by SPSLs.

I have one concern about the inhibition of drugs related to adipocyte differentiation and lipid accumulation by SPSLs. I think that SPSL is like dietary fiber and adsorbs other components such as triglycerides, cholesterol, and bile acids. Therefore, could it be that SPSL did not reduce lipid accumulation in adipocytes and zebrafish, but rather attenuated the effects of drugs that promote fat accumulation? Please discuss this point in the Discussion or Limitation of this study.

Response :

We appreciate your valuable insights. Previous research suggests that dietary fiber possesses notable lipid adsorption properties and decelerates gastrointestinal motility, thereby delaying nutrient absorption. While the possibility of SPSL, as a dietary fiber derived from edible seaweed, exhibiting in vivo lipid absorption cannot be dismissed, reports of general dietary fiber concurrently activating energy metabolism via AMPK-PPAR gamma remain scarce. However, prior studies have documented that fucose derived from edible seaweed induces thermogenesis enhancement-induced energy expenditure in adipose tissue [1]. Should SPSL administration disrupt normal lipogenesis and merely limit fat accumulation, it could potentially trigger adipocyte apoptosis, a phenomenon with numerous documented instances [2-4]. If SPSL functions in this manner, it might have elicited toxicity in our cellular and zebrafish models. However, given the absence of any impact on cell and zebrafish viability, this possibility was not considered. In addition, SPSL-induced the inhibition of lipid accumulation in adipocytes was partially restored by AMPK agonist treatment (Fig. 5). Thus, these result also strongly supports the anti-obesity mechanism of SPSL are not linked to the role of general dietary fibers in physiological conditions.

Nonetheless, it is imperative to acknowledge the inherent limitations of the present study. Notably, the long-term ramifications of SPSL administration in the zebrafish model remain uninvestigated. While the utilization of a widely consumed food material is promising, a comprehensive safety evaluation is warranted to ascertain its suitability as a health functional food ingredient. In our discussion part (line 341-348), we propose that SPSL's beneficial effects on obesity are not merely due to the typical lipid adsorption and excretion mechanisms seen with common dietary fibers. Instead, we suggest, based on inhibitor studies, that SPSL exerts its effects by actively suppressing lipid accumulation through the activation of AMPK pathways. We believe this distinct mechanism of action represents a clear point of differentiation for our material.

Line 340-347 “However, SPSL treatment dose-dependently reversed these patterns, impacting the AMPK pathway as well as PPARγ and its downstream targets for lipogenesis, such as SREBP1 and ATGL (Fig. 4A-G). Previous studies, have demonstrated that fasting or agonist-induced AMPK activation reduces PPARγ transcription in adipose tissues in vivo [5]. This SPSL-induced anti-adipogenic effect was reversed by AMPK inhibition via compound C co-incubation during adipocyte differentiation (Fig. 5A-B). Therefore, SPSL-stimulated AMPK activation in adipocytes likely regulates adipogenesis and lipogenesis.”

Comment 2:

  1. SPSLs is high-molecular weights

The author aims to develop SPSL as a health functional food. If so, the cell experiments in this study seem to be far from the purpose, since SPSL is of high molecular weight and cannot be absorbed at this size. If the authors are committed to this goal, they should describe the limitations of this cell study.

 Response :

We acknowledge the reviewer's astute observation regarding the potential challenge of absorption for a high molecular weight polysaccharide like the one investigated in our study. This is indeed an area that warrants further exploration.

L-fucose-rich sulfated polysaccharides, specifically containing sulfate groups, exhibit high water solubility compared to general dietary fibers presenting gel-forming properties. Also, this suggests a relatively high bioavailability of SPSL. Therefore, the presence of sulfate groups is a key factor distinguishing SPSL from many other dietary fibers in terms of solubility, and this enhanced solubility may indeed contribute to its potentially higher bioavailability.

Despite the high molecular weight of the substance, which may limit its bioavailability, the observed lipid accumulation inhibitory effects in our zebrafish model suggest that SPSL supplementation may act primarily by modulating the gut microbiota composition or its derived metabolite secretion as a prebiotics role.

On your advice we described this point at discussion part,

Line 305-310 “However, the exact mechanism of algal sulfated polysaccharide remain unclear, their abundance of sulfate groups has been linked to high water solubility compared to general dietary fibers presenting gel-forming properties [6]. Therefore, the presence of sulfate groups is a key factor distinguishing SPSL from many other dietary fibers in terms of solubility, and this enhanced solubility may indeed contribute to its potentially higher bioavailability.”

Line 317-334 “Sulfate groups in sulfated polysaccharides, primarily composed of fucoidan, refer to the diverse positions where these groups attach to algal monosaccharides. Fucose, the main monosaccharide in fucoidan, can have sulfate groups attached at various positions, such as the 2nd, 3rd, or 4th carbon [7,8]. This diversity in sulfate group attachment sites creates structural variations in fucoidan, potentially influencing its biological functions, including antioxidant, anti-inflammatory, and anti-cancer effects [9]. Numerous studies have revealed the relationship between fucoidan's structure and specific physiological activities [8,10,11]. While the precise positional elucidation of sulfate isomers within SPSL remains challenging, identifying sulfate groups associated with specific monosaccharides and the fundamental structural unit in SPSL, which exhibits remarkable anti-obesity effects, lays a foundation for future investigations. In summary, notwithstanding its high molecular weight, the abundance of sulfated glycan units within SPSL suggests it possesses favorable aqueous solubility, facilitating its transit through the gastrointestinal tract. Subsequent utilization by digestive enzymes or the gut microbiome may yield diverse sulfate isomers, potentially eliciting a range of physiological effects, including anti-obesity activity. Further investigation is warranted to elucidate the precise mechanisms and therapeutic implications of these potential bioactivities.”

Comment 3:

  1. Abbreviations

AMPK and PPAR in Abstract are being used without the presentation of official names. I do not know the rules of this journal, but if this is the normal rule, the official name and abbreviation should be presented first, and then the abbreviation should be used thereafter.

 Response :

We are deeply appreciated to your thoughtful consideration onto key points of our manuscript. We edited the official name of these transcription factors at the abstract portion.

Line 22-24In vitro analysis demonstrated potent anti-lipogenic properties in adipocytes, mediated by the downregulation of key adipogenic modulators, including 5’ adenosine monophosphate-activated protein kinase (AMPK) and peroxisome proliferator-activated receptor γ (PPARγ) pathways.”

Comment 4:

  1. Title

The “L” of L-fucose is small capital.

 Response :

Thank you for your careful comments. Your input will help us provide more accurate information to our readers. We edited follow your comments. We edited entire keywords following your comments.

Comment 5:

  1. Fig. 6C

Points other than NT are hidden and not visible. Therefore, I think this graph is inappropriate. The author should consider changing this graph to a bar chart.

 Response :

Thank you for your valuable feedback. I believe your insights will significantly improve our data presentation to readers. We replaced the figure in Fig. 6C.

Comment 6:

  1. Statistical analysis

The authors used the T-test as comparison between multiple groups. But, the t-test is a test method used for comparisons between two groups. Thus, please change to another test. T-test is very prone to significant differences, so using the correct method should result in no significant differences in many items. Therefore, it is necessary to use the correct statistical analysis. If the statistical processing method changes, the conclusion will also change.

Response :

We sincerely apologize for any unclear communication regarding our statistical methods. In our research, we employed both One-way ANOVA and unpaired t-tests, depending on the data being analyzed. For instance, in Fig. 2B, we used a T-test for the sulfate content comparison between SLC and SPSL as it involved a 1:1 comparison. For all other statistical analyses, we utilized one-way ANOVA.

On your advice we added the statistical analysis part –

Line 521-526 "The statistical significance of observed differences was assessed through rigorous hypothesis testing. One-way analysis of variance (ANOVA) was employed for multiple group comparisons, while unpaired t-tests were utilized for pairwise comparisons (e.g., Fig. 2B sulfate content). All statistical analyses were conducted using GraphPad Prism 10 software (GraphPad Software Inc., CA, USA).”

References

  1. Zuo, J.; Zhang, Y.; Wu, Y.; Liu, J.; Wu, Q.; Shen, Y.; Jin, L.; Wu, M.; Ma, Z.; Tong, H. Sargassum fusiforme fucoidan ameliorates diet-induced obesity through enhancing thermogenesis of adipose tissues and modulating gut microbiota. International Journal of Biological Macromolecules 2022, 216, 728-740, doi:https://doi.org/10.1016/j.ijbiomac.2022.07.184.
  2. Jeyakumar, S.M.; Vajreswari, A.; Sesikeran, B.; Giridharan, N.V. Vitamin A supplementation induces adipose tissue loss through apoptosis in lean but not in obese rats of the WNIN/Ob strain %J Journal of Molecular Endocrinology. 2005, 35, 391-398, doi:10.1677/jme.1.01838.
  3. Rayalam, S.; Yang, J.Y.; Ambati, S.; Della-Fera, M.A.; Baile, C.A. Resveratrol induces apoptosis and inhibits adipogenesis in 3T3-L1 adipocytes. Phytotherapy research : PTR 2008, 22, 1367-1371, doi:10.1002/ptr.2503.
  4. Wu, L.-Y.; Chen, C.-W.; Chen, L.-K.; Chou, H.-Y.; Chang, C.-L.; Juan, C.-C. Curcumin Attenuates Adipogenesis by Inducing Preadipocyte Apoptosis and Inhibiting Adipocyte Differentiation. Nutrients 2019, 11, 2307, doi:10.3390/nu11102307.
  5. Kajita, K.; Mune, T.; Ikeda, T.; Matsumoto, M.; Uno, Y.; Sugiyama, C.; Matsubara, K.; Morita, H.; Takemura, M.; Seishima, M.; et al. Effect of fasting on PPARgamma and AMPK activity in adipocytes. Diabetes Res Clin Pract 2008, 81, 144-149, doi:10.1016/j.diabres.2008.05.003.
  6. Wang, Y.; Xing, M.; Cao, Q.; Ji, A.; Liang, H.; Song, S. Biological Activities of Fucoidan and the Factors Mediating Its Therapeutic Effects: A Review of Recent Studies. 2019, 17, 183.
  7. Li, B.; Lu, F.; Wei, X.; Zhao, R. Fucoidan: structure and bioactivity. Molecules (Basel, Switzerland) 2008, 13, 1671-1695, doi:10.3390/molecules13081671.
  8. Ale, M.T.; Meyer, A.S. Fucoidans from brown seaweeds: an update on structures, extraction techniques and use of enzymes as tools for structural elucidation. RSC Advances 2013, 3, 8131-8141, doi:10.1039/C3RA23373A.
  9. Shi, L. Bioactivities, isolation and purification methods of polysaccharides from natural products: A review. Int J Biol Macromol 2016, 92, 37-48, doi:10.1016/j.ijbiomac.2016.06.100.
  10. Lee, J.H.; Park, J.E.; Han, J.S. Fucoidan Stimulates Glucose Uptake via the PI3K/AMPK Pathway and Increases Insulin Sensitivity in 3T3-L1 Adipocytes. Journal of Life Science 2021, 31, 1-9.
  11. Liu, Y.; Wang, N.; Tian, Y.; Chang, Y.; Wang, J. Fucoidans from Thelenota ananas with 182.4 kDa Exhibited Optimal Anti-Adipogenic Activities by Modulating the Wnt/β-Catenin Pathway. Journal of Ocean University of China 2021, 20, 921-930, doi:10.1007/s11802-021-4681-8.

Round 2

Reviewer 1 Report

Comments and Suggestions for Authors

Dear the Editor

These authors reasonably provided all responses to the comments raised by this Reviewer.

Reviewer 2 Report

Comments and Suggestions for Authors

I am satisfied with the revisions that have been made by the authors.